

# Risk factors of bloodstream infection in erythroderma from atopic dermatitis, psoriasis, and drug reactions: a retrospective observational cohort study

Qian Liufu[1,*], Lulu Niu[2,*], Shimin He[2,3], Xuejiao Zhang[2] and Mukai Chen[2]

[1] Department of Dermatology, The First Affiliated Hospital of GuangZhou Medical University, Guangzhou, Guangdong, China
[2] Department of Dermatology, The First Affiliated Hospital of Sun Yat-sen University, Guangzhou, Guangdong, China
[3] Department of Dermatology, The Seventh Affiliated Hospital of Sun Yat-sen University, Shenzhen, Guangdong, China
* These authors contributed equally to this work.

Corresponding author
Mukai Chen,
chenmuk@mail.sysu.edu.cn

## ABSTRACT

**Background:** Atopic dermatitis (AD), psoriasis, and drug reactions associated with erythroderma are frequently complicated by infections. However, bloodstream infection (BSI) have received less research attention.

**Objectives:** This study aimed to investigate the clinical characteristics and risk factors associated with BSI in patients with erythroderma.

**Methods:** A retrospective analysis was conducted on 141 erythroderma cases. Eleven cases were identified as having BSI. Clinical records of both BSI and non-BSI groups were reviewed and compared.

**Results:** BSI was diagnosed in 7.80% (11/141) of erythroderma cases, with a breakdown of 7.14% in AD, 2.00% in psoriasis, and 17.14% in drug reactions. Notably, all positive skin cultures (7/7) showed bacterial isolates concordant with blood cultures. Univariate logistic regression analysis revealed several significant associations with BSI, including temperature ($\leq$36.0 or $\geq$38.5 °C; odds ratio (OR) = 28.06; $p < 0.001$), chilling (OR = 22.10; $p < 0.001$), kidney disease (OR = 14.64; $p < 0.001$), etiology of drug reactions (OR = 4.18; $p = 0.03$), albumin (ALB) (OR = 0.86; $p < 0.01$), C-reaction protein (CRP) (OR = 1.01; $p = 0.02$), interleukin 6 (IL-6) (OR = 1.02; $p = 0.02$), and procalcitonin (PCT) (OR = 1.07; $p = 0.03$). Receiver operating characteristic (ROC) curves demonstrated significant associations with ALB ($p < 0.001$; the area under curve (AUC) = 0.80), PCT ($p = 0.009$; AUC = 0.74), and CRP ($p = 0.02$; AUC = 0.71).

**Conclusions:** Increased awareness of BSI risk is essential in erythroderma management. Patients with specific risk factors, such as abnormal body temperature ($\leq$36.0 or $\geq$38.5 °C), chilling sensations, kidney disease, a history of drug reactions, elevated CRP ($\geq$32 mg/L), elevated PCT ($\geq$1.00 ng/ml), and low albumin ($\leq$31.0 g/L), require close monitoring for BSI development.

## INTRODUCTION

Erythroderma, also known as exfoliative dermatitis, is a severe skin condition characterized by widespread redness and scaling affecting more than 90% of the body's surface area (*Miyashiro & Sanches, 2020*; *Khaled, Sellami & Fazaa, 2010*). However, it is not a specific disease itself. Instead, it often arises as a complication of pre-existing skin conditions, such as atopic dermatitis (AD), psoriasis, drug reactions, or cutaneous lymphomas (*Miyashiro & Sanches, 2020*). In rare cases, erythroderma can be idiopathic, meaning no underlying cause can be identified. Unfortunately, precise data regarding its prevalence and incidence are currently unavailable.

Erythroderma is a serious dermatologic emergency requiring hospitalization for immediate evaluation and treatment (*Inamadar & Ragunatha, 2018*; *Rothe, Bernstein & Grant-Kels, 2005*). Investigation of its etiologies is crucial and guides treatment decisions (*Inamadar & Ragunatha, 2018*). Additionally, managing systemic complications significantly impacts both prognosis and healthcare costs. Common systemic manifestations include infections, hyperthermia, hypoalbuminemia, peripheral edema, and tachycardia (*Li & Zheng, 2012*). These patients are at high risk of bacterial infections due to compromised immune function from immunosuppression therapy or steroids (*Li & Zheng, 2012*; *Cesar et al., 2016*). However, administering antibiotics in such cases can be challenging due to various factors that need careful consideration.

Bloodstream infection (BSI) is major concern in medicine, and they are associated with high mortality and increased healthcare costs (*Nielsen et al., 2016*; *Loonen et al., 2014*). They are typically defined by the presence of living microorganisms in the bloodstream, confirmed by a positive blood culture (*Lamy, Sundqvist & Idelevich, 2020*). Blood culture remains the gold standard for diagnosis, despite novel emerging techniques (*Loonen et al., 2014*; *Laupland & Leal, 2020*; *Opota et al., 2015*). Early recognition and timely administration of appropriate antibiotics are crucial for successful treatment.

Patients with compromised skin barriers or chronic skin diseases are known to have an increased risk of developing infections and sepsis (*Pulido-Perez et al., 2021*). Erythroderma, a severe skin condition affecting most of the body and causing a dysfunctional skin barrier, has been rarely studied in relation to BSI complications. This article investigates BSI in patients with erythroderma, aiming to identify risk factors that can help clinicians identify potential BSI cases.

## MATERIALS AND METHODS

### Study population and diagnostic criteria

This study reviewed the medical records of 141 erythroderma patients admitted to the First Affiliated Hospital of Sun Yat-sen University between January 2014 and December 2019.

These patients were categorized as those with AD, psoriasis, or drug reactions based on their medical history and dermatological diagnosis. The study was approved by the medical ethics committee of the hospital (No. (2021)428). As this was a retrospective study, informed consent from individual patients was not required. Therefore, the requirement for informed consent was waived by the IEC for Clinical Research and Animal Trials at the First Affiliated Hospital of Sun Yat-sen University before collecting patients' medical data.

We reviewed patients' medical history, clinical manifestations, laboratory examinations, and microbial test results from blood cultures and skin swab cultures. Blood samples and skin swabs were both baseline samples. The diagnosis of BSI followed the criteria established by the US Centers for Disease Control and Prevention/National Healthcare Safety Network (CDC/NHSN) surveillance system. These criteria include isolation of a recognized pathogen, which does not include organisms that are considered common skin contaminants, from at least one blood culture and the presence of at least one of the following signs or symptoms: fever (>38 °C), chills, or hypotension.

Patients with blood cultures yielding only contaminants were excluded (*Horan, Andrus & Dudeck, 2008*). Patients diagnosed with BSI were classified as the BSI group, while those with negative or contaminant blood cultures were classified as the non-BSI group.

### Statistical analysis

In this study, different statistical methods were used to analyze the data depending on its type. Normally distributed data were analyzed by utilizing the mean ± standard deviation (SD), the median, and interquartile range (IQR) for skewed data, and percentages were employed to represent categorical data. To compare the statistical differences between the two groups, the unpaired Student's *t* test was employed for normally distributed data, while the Mann–Whitney U test was used for skewed data. Fisher's exact test was used to compare categorical data existing between the two groups. We also applied univariate logistic regression analysis to investigate factors associated with BSI. The covariates considered included temperature, chilling, white blood cell (WBC), albumin (ALB), C-reaction protein (CRP), interleukin 6 (IL-6), procalcitonin (PCT), erythrocyte sedimentation rate (ESR), neutrophil to lymphocyte ratio (N/L ratio), lactic dehydrogenase (LDH), age, sex, kidney disease, diabetes, history of steroid, etiologies of drug reactions, psoriasis, and AD. We performed the Chi-square test to check the dependency between temperature and chilling. Multivariate logistic regression was also performed. The continuous data were analyzed by baseline samples. Receiver operating characteristic (ROC) curves were used to identify the best cut-off values for predicting BSI based on different factors. The cut-off points of factors from ROC curves yielded the high sensitivity and specificity to predict BSI. All statistical tests of the hypothesis were two-sided and performed at the 0.05 level of significance. Statistical analyses were performed using SPSS statistic software (version 23.0, IBM, lnc, Armonk, US). Graphs were created with GraphPad Prism software (version 9.1.2; GraphPad Software, lnc, La Jolla, CA, USA).

## RESULTS

### Demography and clinical manifestations

In this cohort, we included 141 erythroderma cases, categorized as AD (56 cases), psoriasis (50 cases), and drug reactions (35 cases). Among these, 11 cases (7.80%) were diagnosed with BSI (≥1 positive blood culture result, excluding contaminations) (Table 1). The BSI rates across different causes of erythroderma were 7.14% in AD (4/56), 2.00% in psoriasis (1/50), and 17.14% in drug reactions (6/35) (Fig. 1A). The incidence of BSI in drug reactions was significantly higher than that in psoriasis ($p = 0.018$). In these cases, skin swabs/catheter tip cultures were positive in 63.6% (7/11). Also, the bacterial isolates of positive cultures showed 100% (7/7) concordance with blood cultures (Table 1).

### Frequency of bacterial isolates in blood cultures and skin swab cultures

In blood culture isolates (Fig. 1B), seven cases were *Staphylococcus aureus* (*S. aureus*), including five cases of methicillin-resistant *Staphylococcus aureus* (MRSA). Besides, there were two cases of methicillin-resistance coagulase-negative staphylococcus (MRSCON), one case of *Staphylococcus epidermis* (*S. epidermis*), and two cases of *Corynebacterium striatum* (*C. striatum*). As for the bacterial isolates of clinical interest in skin swab cultures (normal colonization bacteria, such as *S. epidermis*, not included), the frequency of bacterial isolates between these etiologies was different (Fig. 1C). In AD, the positive rate was 95.6% (22/23), with 18 (81.8%) isolates of *S. aureus* (6 of MRSA) and four isolates of *Staphylococcus hemolyticus* (*S. hemolyticus*) (2 of MRSCON). In psoriasis, the positive rate was 56.25% (9/16). Among them, there were seven isolates of *S. aureus* (2 of MRSA), two isolates of MRSCON, two isolates of *Escherichia coli* (*E. coli*), one isolate of *Proteus mirabilis* (*P. mirabilis*), and one isolate of *Enterococcus faecium* (*E. faecium*). In drug reactions, 50% (5/10) was positive, with four isolates of *S. aureus* (1 of MRSA), one isolate of MRSCON, one isolate of *Enterococcus faecalis* (*E. faecalis*), one isolate of *P. mirabilis*, and one isolate of Klebsiella pneumonia (*K. pneumonia*).

### Characteristics and laboratory examination of BSI

Next, we compared the differences between the patients with BSI and those without BSI (Fig. 2). For incidences of BSI, there were significant differences in groups classified by temperature (34.60% *vs.* 1.85%, $p < 0.001$), history of steroids (13.46% *vs.* 4.49%, $p = 0.01$), chilling (36.36% *vs.* 2.52%, $p < 0.001$), and kidney disease (41.67% *vs.* 4.65%, $p = 0.001$). No significant differences in incidence were found in the subgroup of diabetes and sex. However, comparing the BSI group with the non-BSI group, we also found that there were significant differences in CRP (median (IQR): 14.73 mg/L (5.06–44.00 mg/L) *vs.* 39.00 mg/L (15.13–164.00 mg/L), $p = 0.02$), ESR (mean ± SD: 26.48 ± 22.93 mm/h *vs.* 47.83 ± 41.30 mm/h, $p = 0.04$), PCT (median (IQR): 0.10 ng/mL (0.05–0.33 ng/mL) *vs.* 0.48 ng/mL (0.10–5.23 ng/mL), $p = 0.006$), and ALB (median (IQR): 34.95 g/L (31.00–37.03 g/L) *vs.* 28.00 g/L (26.00–31.00 g/L), $p < 0.001$). No significant differences were found in WBC counts, N/L ratio, IL-6, and LDH. More detailed were showed in Table S1.

**Table 1 Demographic and clinical details of cases with bloodstream infection in the context of erythroderma from atopic dermatitis, psoriasis and drug reactions.**

| No | Etiology | Blood culture | Skin swabs culture/ Catheter tips culture S.aureus | Chilling | Heart rate (per min) | Temperature (°C) | Blood pressure (mmHg) | History of steroid | Therapy of etiology | Treatment of infection |
|---|---|---|---|---|---|---|---|---|---|---|
| 1 | Drug reactions | *Two positives: S.aureus* | S.aureus (skin) | Yes | 92 | 37.8 | 116/65 | Yes | Steroid | Linezolid, Ceftazidime, |
| 2 | Drug reactions | *One positive: S. epidermis* | Negative (skin) | Yes | 130 | 39.5 | 120/80 | Yes | Steroid, Immunoglobulin | Ceftazidime, Teicoplanin |
| 3 | Atopic dermatitis | *One positive: MRSA* | MRSA (skin) | Yes | 116 | 40.4 | 158/83 | Yes | Steroid, Antihistamine, Topical Steroid | Meropenem, Levofloxacin |
| 4 | Atopic dermatitis | *One positive: S. aureus* | S.aureus (skin) | No | 110 | 38.3 | 119/72 | No | Steroid, Antihistamine, Topical Steroid | Cefoperazone, Levofloxacin |
| 5 | Drug reactions | *Two positives: MRSCON; two positives: C.striatum* | C. striatum (catheters tips) | Yes | 104 | 36 | 136/95 | No | Steroid, Immunoglobulin | Imipenem, Levofloxacin, Azithromycin |
| 6 | Atopic dermatitis | *One positive: MRSA* | Negative (skin) | No | 96 | 38.7 | 142/87 | No | Antihistamine, Hydroxychloroquine, Topical steroid | Linezolid, Clindamycin |
| 7 | Psoriasis | *One positive: MRSCON; one positive: MRSA(from other institution)* | MRSCON; EBSL; E.faecium; P.mirabilis (skin) | Yes | 92 | 40 | 120/75 | Yes | Steroid, Methotrexate, Cyclosporin A, Topical steroid | Vancomycin, Imipenem |
| 8 | Atopic dermatitis | *Three positives: MRSA* | MRSA (skin) | No | 103 | 38.6 | 118/71 | No | Antihistamine, Topical Steroid | Linezolid, Ceftazidime, Teicoplanin, |
| 9 | Drug reactions | *One positive: C.striatum* | negative (skin) | Yes | 130 | 40 | 102/68 | Yes | Antihistamine | Teicoplanin |
| 10 | Drug reactions | *Two positives: MRSA* | MRSA; K. pneumoniae (skin) | Yes | 95 | 38.6 | 166/71 | Yes | Steroid, Immunoglobulin, Cyclosporin A, Antihistamine | Linezolid |
| 11 | Drug reactions | *Two positives: MRSA* | Negative (skin) | Yes | 125 | 39.9 | 182/73 | Yes | Steroid, Immunoglobin | Linezolid, Vancomycin |

**Note:**
*S. aureus, Staphylococcus aureus*; *S. epidermis, Staphylococcus epidermis*; MRSA, Methicillin-resistance staphylococcus aureus; *C. striatum, Corynebacterium striatum*; MRSCON, Methicillin-resistant coagulase-negative staphylococcus; *E. faecium, Enterococcus faecium*; *P.mirabilis, Proteus mirabilis*;*K. pneumonia, Klebsiella pneumonia*.

## Risk factors of BSI

To identify the risk factors for BSI, we compared the BSI and non-BSI groups and analyzed various factors using univariate logistic regression (Fig. 3). For the categorical covariables, there were significant associations of temperature (≤36.0 or ≥38.5 °C), chilling, kidney disease, and history of drug reactions with BSI. However, there were no significant

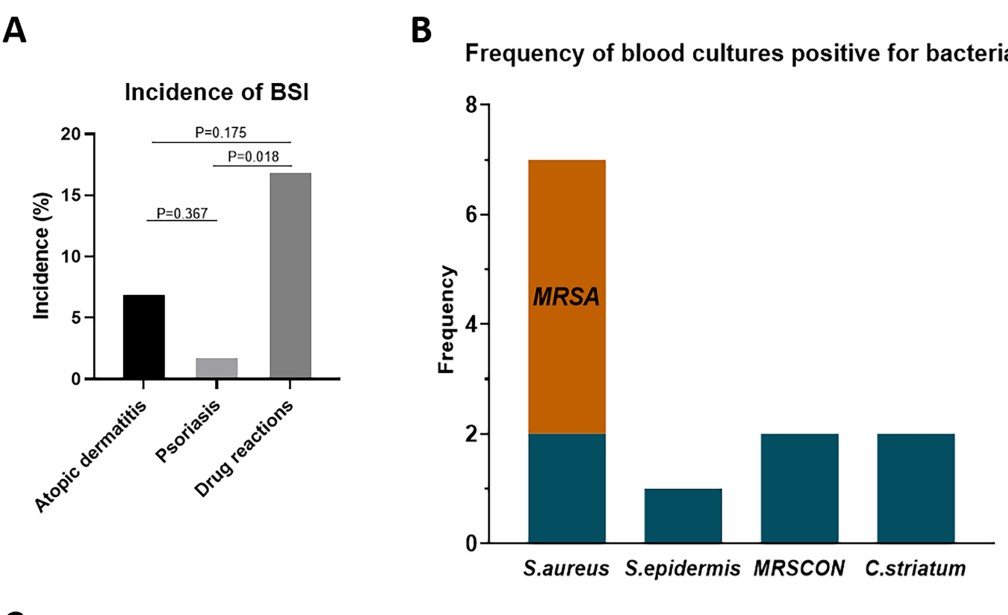

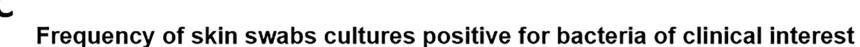

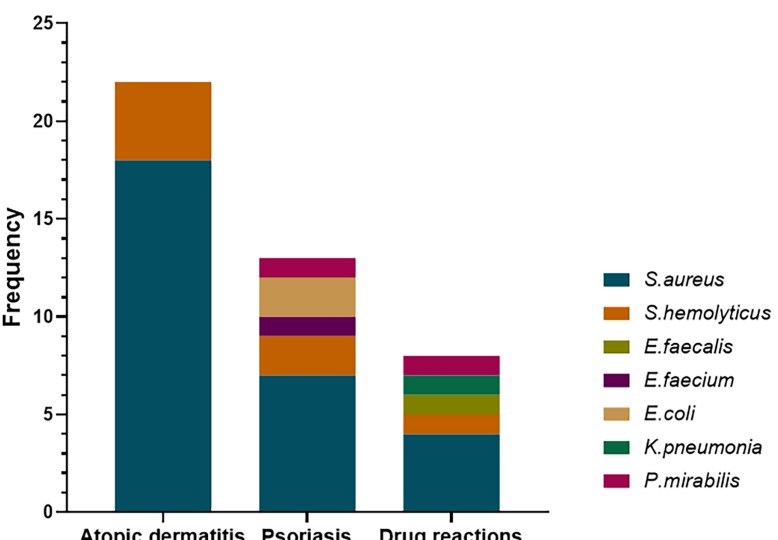

**Figure 1 Incidence of BSI in erythroderma and frequency of bacterial isolates from blood cultures and skin swab cultures.** (A) Incidence of BSI in patients with erythroderma due to atopic dermatitis, psoriasis, and drug reactions. (B) Frequency of isolates from blood cultures positive for bacteria. (C) Frequency of isolates from skin swab cultures positive for bacteria of clinical interest. BSI, bloodstream infection; *S. aureus*, staphylococcus aureus; *S. epidermis*, staphylococcus epidermis; *S. hemolyticus*, staphylococcus hemolyticus; *E. faecalis*, enterococcus faecalis; *E. faecium*, enterococcus faecium; *K. pneumonia*, klebsiella pneumonia; *P. mirabilis*, proteus mirabilis; MRSA, methicillin-resistance staphylococcus aureus; MRSCON, methicillin-resistant coagulase-negative staphylococcus.

associations with sex (male), history of steroids, diabetes, etiology of AD, and etiology of psoriasis, as depicted in Fig. 3A. For continuous covariables, there were significant associations of ALB, CRP, IL-6, and PCT with BSI but none for age, LDH, and N/L ratio, as

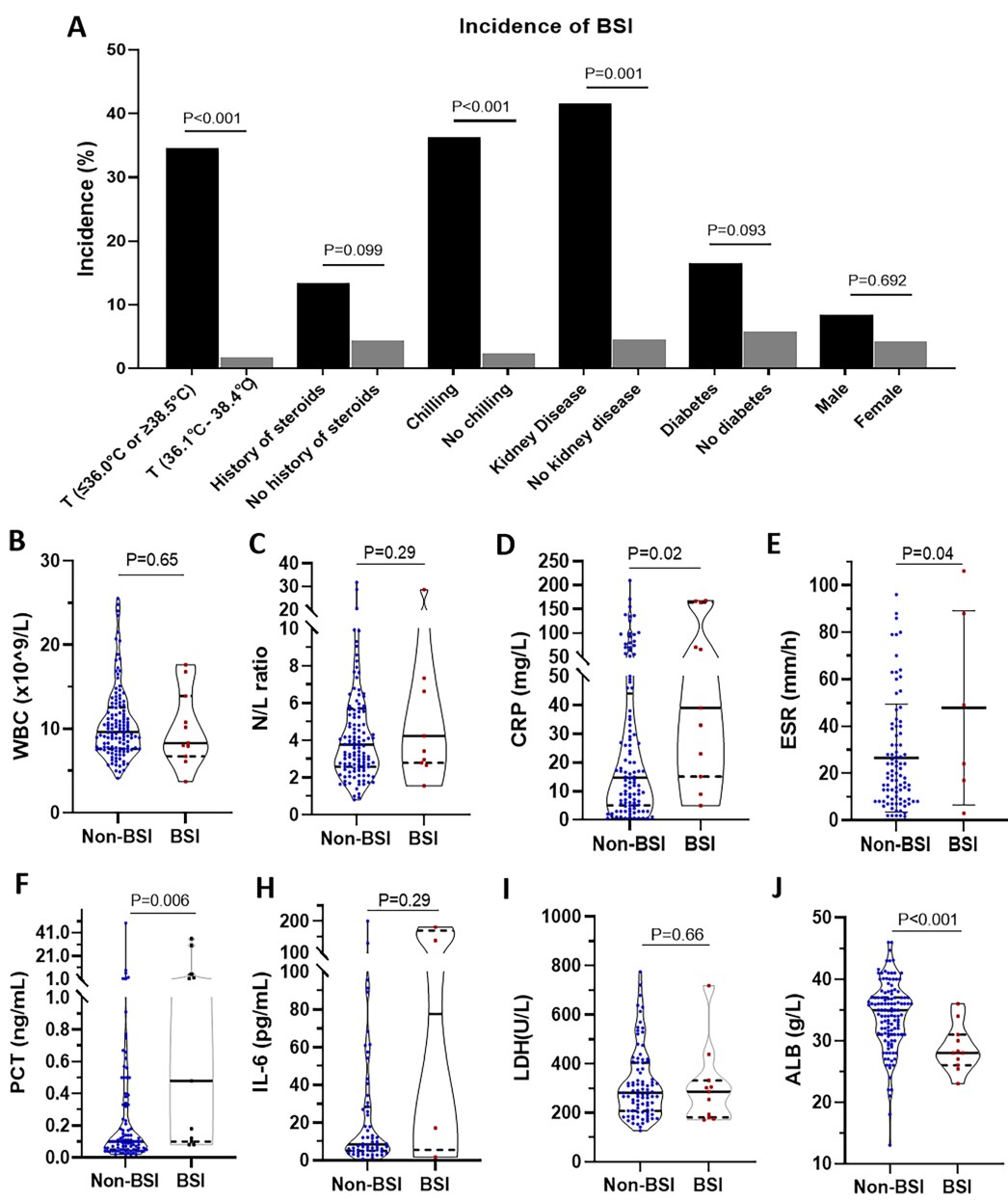

**Figure 2 Comparison of manifestations, clinical history, and laboratory examination between erythroderma patients with bloodstream infection (BSI) and those without BSI.** (A) The incidence of BSI in different groups is classified by temperature, history of steroids, chilling, kidney disease, diabetes, and sex. (B) WBC counts. (C) Neutrophil to lymphocyte ratio (N/L ratio). (D) CRP. (E) ESR. (F) PCT. (G) IL-6. (H) LDH. (I) ALB.           

shown in Fig. 3B (present with OR per unit of factors). We used ROC curves to identify the most relevant continuous factors for predicting BSI in erythroderma (Fig. 3C). More detailed were showed in Table S2. There was significant dependency between temperature and chilling in the Chi-square test (Table S3). Multivariate logistic regression analysis showed that temperature (≤36.0 or ≥38.5 °C) was significant (Table S4). The ROC curves for ALB, PCT, and CRP were statistically significant.

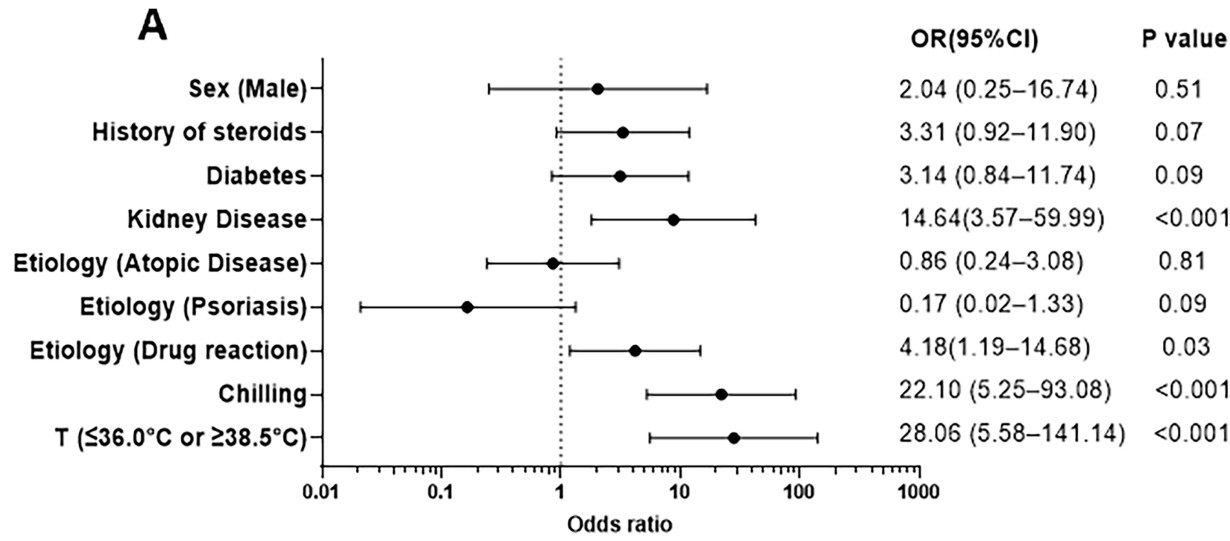

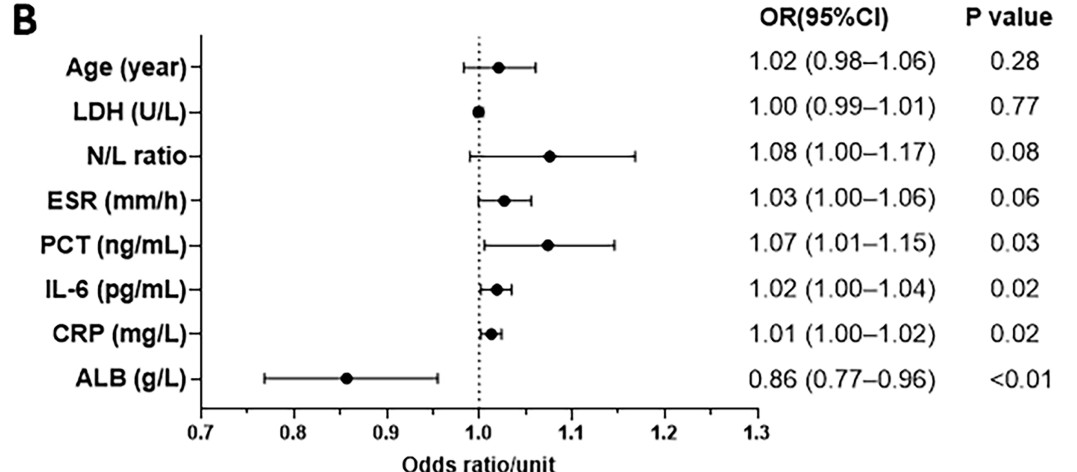

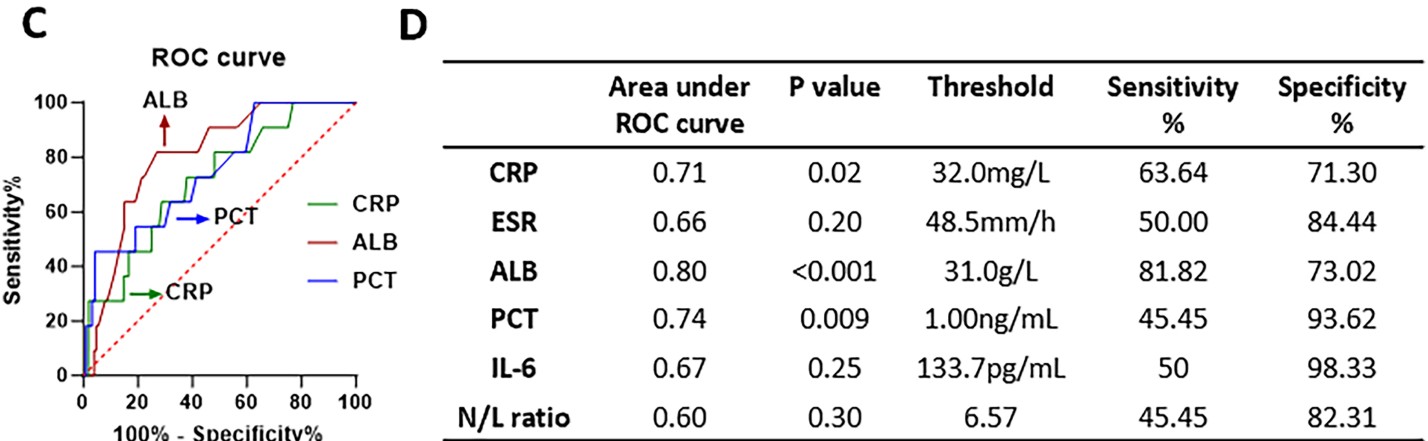

**Figure 3  Odds ratios (ORs) of factors to bloodstream infection (BSI) from univariable logistic regression and receiver operating characteristic (ROC) curve.** (A) OR of factors in logistic regression, for categorical covariables. (B) OR per unit of factors in logistic regression, for continuous covariables. (C) ROC curves and the parameters of some factors.             

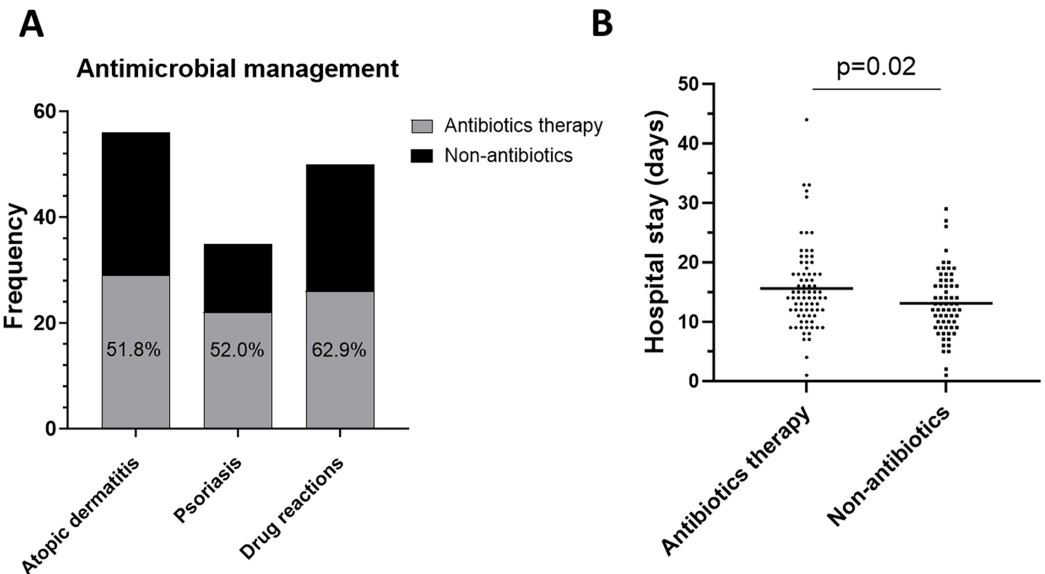

**Figure 4 Antibiotics therapy.** (A) Frequency of antibiotics therapy in atopic dermatitis, psoriasis, drug reactions. (B) Comparison of hospital stay days between with antibiotics and without antibiotics.

## Antibiotics therapy

Antibiotics are commonly used in erythroderma to prevent or control infections. In this study, 54.6% (77/141) of patients with erythroderma received antibiotic therapy (Fig. 4A). Interestingly, the analysis of hospital stay revealed that patients who received antibiotics had a longer stay than those who did not (median (IQR): 14.00 days (11.00–18.00 days) *vs.* 12.50 days (9.00–16.75 days), $p = 0.02$) (Fig. 4B).

## DISCUSSION

To date, limited research exists on BSI in erythroderma. This study found a higher BSI rate (7.80%) in patients with erythroderma from AD, psoriasis, or drug reactions compared to the overall incidence of bacteremia in hospitalization, given as 1.42% (*Nielsen et al., 2016*). This might be due to the large area of impaired skin barrier in erythroderma.

In our study, *S. aureus* bloodstream infection (SAB) was the most common (seven isolates from 11 positive samples), with a high rate of MRSA (5/7). *S. aureus* is by far the most common cause of persistent bacteremia, and it has been associated with increased mortality (*Kuehl et al., 2020*). However, some normal colonized bacteria of the skin (such as *S. epidermidis* and *C. striatum*) also need to be considered, especially in patients with intravascular catheters and in patients on high dose steroid or immunosuppression therapy (*Kleinschmidt et al., 2015*).

Skin swabs/catheter tip cultures from patients with positive blood cultures all showed concordant bacterial isolates (100%, 7/7). It has been reported that skin cultures help predict the pathogen in BSI (*Drinka, Bonham & Crnich, 2012*; *Lecadet et al., 2019*; *Mathé et al., 2020*). Our results also emphasize the importance of monitoring skin colonization

through repeated skin cultures. This may aid in identifying the bacteria most likely involved in BSI before blood culture results become available.

Within the analysis of etiologies, we found that drug reactions had higher morbidity of BSI (17.1%) than psoriasis (2.00%) and AD (7.14%). This might indicate that different conditions of etiologies (such as skin microbiota, skin barrier functions, and innate defense antimicrobial peptides) influence the morbidity of BSI. In AD, all the isolates of blood cultures were *S. aureus*, which corresponded with the results of skin cultures. Studies have shown that 70–90% of patients with AD may have high levels of *S. aureus* colonization, compared to only 20% in the general population (*Mathé et al., 2020*; *Schlievert et al., 2010*; *Totte, van der Feltz & Hennekam, 2016*). Additionally, AD is associated with an increased risk of systemic infections and SAB (*Mathé et al., 2020*; *Serrano, Patel & Silverberg, 2019*; *Droitcourt et al., 2021*). The underlying mechanisms may be a combination of reduced expression of antimicrobial peptides, dysbiosis, dense colonization with *S. aureus*, and impaired skin barrier (*Mathé et al., 2020*; *Schlievert et al., 2010*; *Harkins, McAleer & Bennett, 2018*). Drug reactions expressed significantly higher morbidity of BSI than psoriasis. It was reported in a study that 46.9% of patients with epidermal necrolysis (overlap syndrome or toxic epidermal necrolysis) experienced at least one BSI episode, with skin cultures showing a high degree of concordance (71.1%) with blood cultures (*Lecadet et al., 2019*). The mechanisms that bring about a high risk of BSI in drug reactions are not fully understood, but they may be related to the sudden disruption of the skin barrier caused by necrosis of keratinocytes. Psoriasis has been less reported with regard to infectious complications. This is consistent with the low morbidity of psoriasis-related BSI in our study. It may be related to increased antimicrobial peptides and imbalanced Th1/Th17 response in psoriasis (*Zeeuwen et al., 2013*).

Early diagnosis and treatment of BSI are crucial for better outcomes in erythroderma. However, the timely diagnosis of BSI arising from erythroderma in patients can be challenging in a clinical setting due to the overlap of symptoms, such as hyperthermia, tachycardia, and increased inflammatory indicators (such as CRP, ESR, and WBC). This study aimed to identify the key differences and risk factors that can help clinicians distinguish between erythroderma with and without BSI.

In our study, temperature (≤36.0 or ≥38.5 °C) and chilling had high values of OR (28.06 and 22.10, respectively) in the analysis of univariate logistic regression. This means that they could be good predictors of BSI. Notably, patients with kidney disease or with the etiology of drug reactions also have significantly higher risks for BSI. CRP and PCT, which have been reported to assist in early prediction of BSI (*Doerflinger, Haeusler & Li-Wai-Suen, 2021*; *Pierrakos & Vincent, 2010*), could also be significant predictors, with cut-off points of 32 mg/L and 1.00 ng/ml, respectively, as observed in this study. Standard CRP (0–10 mg/ml) and PCT (0–0.05 ng/ml) levels may not be reliable indicators of BSI in erythroderma patients. This is because our study found a high percentage of erythroderma patients with elevated CRP (61.3%) and PCT (72.9%) even when they did not have BSI. Aside from the common inflammatory indicators, such as CRP and PCT, ALB was also negatively associated with BSI, having an OR of 0.86 ($p < 0.01$). This indicates that hypoalbuminemia (≤31.0 g/L) could be a risk factor for BSI in erythroderma. Interestingly,

it has been reported that hypoalbuminemia could occur as a surrogate and culprit infection (*Wiedermann, 2021*).

Antibiotic therapy is often adopted in the management of erythroderma (*Chao, Wang & Hsu, 2020*). In this study, 54.6% of patients used not less than one antibiotic, and we found that this led to them having longer hospital stays. The decision to adopt antibiotics depends on the evaluation of the risk of infections, especially in the presence of BSI. According to our results, abnormal temperature ($\leq$36.0 or $\geq$38.5 °C), chilling, kidney disease, the etiology of drug reactions, elevated CRP ($\geq$32 mg/L) and PCT ($\geq$1.00 ng/ml), and low ALB ($\leq$31.0 g/L) could be fingered as the predictors of BSI. Antibiotics should be prescribed immediately for patients suspected of having invasive bacterial infections (*Kern & Rieg, 2020*). Importantly, proper selection and dosing of antimicrobial regimens are also critical for favorable outcomes (*Gotts & Matthay, 2016*; *Huttunen et al., 2013*; *Martinez & Wolk, 2016*). Therefore, since skin swab/catheter tip cultures often grow the same bacteria as blood cultures (100% concordance in our study), targeting antibiotics toward these skin culture results may be a valuable strategy (*Lecadet et al., 2019*). An often overlooked but essential line of action is to promptly carry out a blood culture when BSI is suspected (*Fabre, Sharara & Salinas, 2020*; *Khatib, Riederer & Saeed, 2005*). If the situation does not improve, repeated blood cultures are necessary.

This study has a number of limitations, most notably the imbalance in numbers between BSI and non-BSI groups due to the rare presentation of erythroderma in the clinic. Despite this limitation, this study may provide shed light on the nature of erythroderma and BSI in the clinic.

## CONCLUSIONS

In conclusion, BSI should be taken seriously in patients with erythroderma, especially in cases with risk factors such as abnormal temperature ($\leq$36.0 or $\geq$38.5 °C), chilling, kidney disease, the etiology of drug reactions, elevated CRP ($\geq$32 mg/L) and PCT ($\geq$1.00 ng/ml), and low ALB ($\leq$31.0 g/L). Antibiotic therapy should be evaluated and may be considered to target the bacteria of skin cultures. Repeated skin cultures and blood cultures are necessary if BSI is suspected.

### Funding
The APC was supported by the Guangzhou Municipal Science and Technology Bureau Project (grant number 2023A04J1863). The funders had no role in study design, data collection and analysis, decision to publish, or preparation of the manuscript.

### Grant Disclosures
The following grant information was disclosed by the authors:
Guangzhou Municipal Science and Technology Bureau Project: 2023A04J1863.

## Competing Interests

The authors declare that they have no competing interests.

## Author Contributions

- Qian Liufu conceived and designed the experiments, performed the experiments, analyzed the data, prepared figures and/or tables, authored or reviewed drafts of the article, and approved the final draft.
- Lulu Niu conceived and designed the experiments, performed the experiments, analyzed the data, prepared figures and/or tables, authored or reviewed drafts of the article, and approved the final draft.
- Shimin He performed the experiments, prepared figures and/or tables, and approved the final draft.
- Xuejiao Zhang performed the experiments, prepared figures and/or tables, and approved the final draft.
- Mukai Chen conceived and designed the experiments, authored or reviewed drafts of the article, and approved the final draft.

## Human Ethics

The following information was supplied relating to ethical approvals (*i.e.*, approving body and any reference numbers):

IEC for Clinical Research and Animal Trials of the First Affiliated Hospital of Sun Yat-sen University.

## Data Availability

The data is available in the Supplemental Files.

## Supplemental Information

Supplemental information for this article can be found online at http://dx.doi.org/10.7717/peerj.17701#supplemental-information.

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
