# Peer review of "Risk factors of bloodstream infection in erythroderma from atopic dermatitis, psoriasis, and drug reactions: a retrospective observational cohort study"

_PeerJ, doi:10.7717/peerj.17701_

## Round 0.1 · original submission · Major Revisions

I have completed my evaluation of your manuscript. The reviewers recommend reconsideration of your manuscript following major revision. I invite you to resubmit your manuscript after addressing the comments below. When revising your manuscript, please consider all issues mentioned in the reviewers' comments carefully: please outline every change made in response to their comments and provide suitable rebuttals for any comments not addressed. Please note that your revised submission may need to be re-reviewed.

**Language Note:** The review process has identified that the English language must be improved. PeerJ can provide language editing services - please contact us at [email protected] for pricing (be sure to provide your manuscript number and title). Alternatively, you should make your own arrangements to improve the language quality and provide details in your response letter. – PeerJ Staff

Reviewer 1 ·

Basic reporting

- Standard of English is mixed. I would recommend the use of a professional English language editing service as there are too many instances of inappropriate grammar for this reviewer to highlight. The overall message is clear, so I commend the authors on that, but it is not currently of sufficient standard for publication.

- The literature citation is appropriate throughout.

- The raw data is available and operable. The article is well structured.

Experimental design

- The research question is well defined and addresses a knowledge gap in the frequency of BSI in erythroderma

- The n numbers are relatively low for some of the analysis performed. Ideally an additional site would be analysed and included to strengthen the claims and bolster the n numbers for each of the conditions studied that give rise to erythroderma. However, the authors have been measured in their claims and not over-interpreted the results so this may not be deemed necessary. The inclusion of a statement of limitations in the Discussion section citing the low n numbers would be helpful in this instance.

- The work appears to be ethically sound.

- The methods are well described and the data is available to perform corroborating studies if required.

Validity of the findings

- The data has been provided in an operable format.

- The conclusions are generally well stated and reflect the results, however the n numbers are low and it is a single centre study which would be greatly strengthened by the inclusion of an additional hospital centre.

Reviewer 2 ·

Basic reporting

In this paper, Qian et al. conducted a retrospective cohort study to investigate the risk factors of bloodstream infections in erythroderma. Among 141 erythroderma, 11 are BSI. Temperature, chilling, kidney disease, etiology of drug reactions, ALB, CRP, PCT were identified as risk factors for BSI. The manuscript is structured well. However, there are some general comments:
1. The language should be polished. There are some grammar issues and typos, such as 1) line 37, "is a", "a" should be deleted. 2) line 152, "we found that drug reactions had higher morbidity (17.1%) than psoriasis (2.00%) and AD (7.14%)." I think the authors want to express the morbidity of BSI in different subgroups. Please modify the language to avoid confusion.

2. I'm not sure if the author's names are presented consistently. Please double check.

3. It is better to start with capitalized letters for the titles of figures and tables. And I would suggest the author add a figure legend to Figure 1B.

Experimental design

1. Are blood samples and skin swabs baseline samples? I think the authors should clarify this in the manuscript. The same comment applies to the covariates.

2. Could authors provide a summary table for demographic characteristics, which is essential and more straightforward to get an idea of the cohort and difference between BSI/non-BSI?

Validity of the findings

1. Comparisons are performed univariately. Is there any confounding effect for each comparison, and if so, how is confounding controlled? In addition, maybe there is some overlap among the covariates? To me, temperature and chilling are correlated and reflect the same thing from different perspectives.

2. What are the sensitivity and specificity of the cutoffs selected for ALB, PCT and CRP? What statistical test is used to calculate the p-value of AUC? It seems this is not mentioned in the method part.

3. A major limitation of the findings is that the prevalence of BSI is only 7.8% of the 141 samples. BSI and non-BSI are very imbalanced in each comparison, which makes the results underpowered. The authors should discuss this limitation and other limitations.

Reviewer 3 ·

Basic reporting

The article was well-written and was easy to follow throughout the manuscript. The authors asked a specific question in this manuscript on "whether BSI predictors could be established for erythroderma patients." They structured their paper to answer their specific question and provided relevant data to make their conclusions.

Experimental design

The study was done in a retrospective data set therefore the data pool was small for drawing significant conclusions. The authors have used appropriate statistical analysis to determine association and ROC findings to determine if any comparators are associated with the probability of a patient getting a BSI.

Validity of the findings

Although the temperature has shown a significant association in BSI patients versus non-BSI patients, it was interesting to notice that the authors used 38.4 degrees Celsius (101F) as the upper limit, considered fever in many hospitals.
Another interesting finding was that BSI patients receiving antibiotics stayed longer in the hospital than patients who did not receive antibiotics. BSI infections are known to have high mortality and assuming these patients did not die due to their BSI infection (which was not mentioned in the manuscript), the question then arises whether antibiotics are impacting the recovery of the BSI patients. I want the author to address this point as the main point of this manuscript is "When will it be ideal to prescribe antibiotics for erythroderma patients being affected by BSI."

---

## Round 0.2 · Minor Revisions

I have completed my evaluation of your manuscript. The reviewers recommend reconsideration of your manuscript following minor revision. I invite you to resubmit your manuscript after addressing the comments below. When revising your manuscript, please consider all issues mentioned in the reviewers' comments carefully: please outline every change made in response to their comments and provide suitable rebuttals for any comments not addressed. Please note that your revised submission may need to be re-reviewed.

Reviewer 1 ·

Basic reporting

The quality of language has improved to a publishable standard, though is til ambigious in places.

For instance the additional line in the discussion (L212-214):
"This study had limitations that the included cases were limited, because erythroderma is a rare condition in clinic. This study could provide some clues when there were some confusions about BSI of erythroderma in clinic."

This is not clear.

I would suggest something like:
"This study has a number of limitations, most notably the imbalance in n numbers between BSI and non-BSI groups due to the rare presentation of erythroderma in the clinic. Despite this limitation, this study may provide shed light on the nature of erythroderma and BSI in the clinic."

Experimental design

I am satisfied that the authors have addressed all the reviewers comments within reason.

Validity of the findings

I refer to my section 1 statement around acknowledgements of the study's limitations and suggest altering L212-214

Reviewer 2 ·

Basic reporting

The manuscript has been substantially polished. I just have a minor comment: it should be "univariate logistic regression," not "univariable" logistic regression.

Experimental design

No comment

Validity of the findings

I'd like to clarify my previous comments. My previous comment was that the authors may consider fitting multivariate logistic models between BSI and the covariates that they studied in the univariate analysis to identify risk factors in order to control potential confounding effects. If checked independently, some covariates may not be the true "risk factors" because they are actually driven by other covariates. On the other hand, the dependency between temperature and chilling can be checked by the Chi-square test. If significantly dependent, only one of them should be included in the multivariate model, otherwise the results may be unstable due to the multicollinearity. I would suggest the authors check the above two comments again to make the results more reliable.

Reviewer 3 ·

Basic reporting

The authors have done a good job in improving the English and structure of the paper. It was a smooth read.

Experimental design

I have no follow-up questions regarding the experimental design and I am satisfied with their response in the cover letter

Validity of the findings

no comment

---

## Round 0.3 · accepted · Accept

It is a pleasure to accept your manuscript entitled " Risk factors of bloodstream infection in erythroderma from atopic dermatitis, psoriasis, and drug reactions: a retrospective observational cohort study" in its current form for publication in PeerJ.